# SCELMo: Source Code Embeddings from Language Models

## Abstract

Continuous embeddings of tokens in computer programs have been used to support a variety of software development tools, including readability, code search, and program repair. Contextual embeddings are common in natural language processing but have not been previously applied in software engineering. We introduce a new set of deep contextualized word representations for computer programs based on language models. We train a set of embeddings using the ELMo (embeddings from language models) framework of Peters et al (2018). We investigate whether these embeddings are effective when fine-tuned for the downstream task of bug detection. We show that even a low-dimensional embedding trained on a relatively small corpus of programs can improve a state-of-the-art machine learning system for bug detection.

## 1 Introduction

Learning rich representations for source code is an open problem that has the potential to enable software engineering and development tools. Some work on machine learning for source code has used hand engineered features (Long & Rinard, 2016, e.g.), but designing and implementing such features can be tedious and error-prone. For this reason, other work considers the task of learning a representation of source code from data (Allamanis et al., 2018a). Many models of source code are based on learned representations called embeddings, which transform words into a continuous vector space (Mikolov et al., 2013). Currently in software engineering (SE) researchers have used static embeddings (Harer et al., 2018; White et al., 2019; Pradel & Sen, 2018), which map a word to the same vector regardless of its context. However, recent work in natural language processing (NLP) has found that contextual embeddings can lead to better performance (Peters et al., 2018; Devlin et al., 2018; Yang et al., 2019; Liu et al., 2019). Contextualized embeddings assign a different vector to a word based on the context it is used. For NLP this has the advantage that it can model phenomena like polysemy. A natural question to ask is if these methods would also be beneficial for learning better SE representations.

In this paper, we introduce a new set of contextual embeddings for source code. Contextual embeddings have several potential modelling advantages that are specifically suited to modelling source code:

- Surrounding names contain important information about an identifier. For example, for a variable name, surrounding tokens might include functions that take that variable as an argument or assignments to the variable. These tokens provide indirect information about possible values the variable could take, and so should affect its representation. Even keywords can have very different meanings based on their context. For instance, a private function is not the same as a private variable or a private class (in the case of Java / C++).

- Contextual embeddings assign a different representation to a variable each time it is used in the program. By doing this, they can potentially capture how a variable's value evolves through the program execution.

- Contextual embeddings enable the use of transfer learning. Pre-training a large neural language model and querying it for contextualized representations while simultaneously fine-tuning for the specific task is a very effective technique for supervised tasks for which there is a small amount of supervised data available. As a result only a small model needs to be fine-tuned atop the pre-trained model, without the need for task-specific architectures nor the need of training a large model for each task separately.

In this paper, we highlight the potential of contextual code embeddings for program repair. Automatically finding bugs in code is an important open problem in SE. Even simple bugs can be hard to spot and repair. A promising approach to this end is name-based bug detection, introduced by DeepBugs (Pradel & Sen, 2018). The current state-of-the-art in name-based bug detection relies on static representations from Word2Vec (Mikolov et al., 2013) to learn a classifier that distinguishes correct from incorrect code for a specific bug pattern. We introduce a new set of contextualized

embeddings for code and explore its usefulness on the task of name-based bug detection. Our method significantly outperforms DeepBugs as well as other static representations methods on both the DeepBugs dataset as well as a new previously unused test set of JavaScript projects. We release our implementation and representations as they could lead to improvements in a great variety of SE tasks.

## 2 RELATED WORK

Unsupervised static word embeddings have been extensively used to improve the accuracy of supervised tasks in NLP (Turian et al., 2010). Notable examples of such methods are Word2Vec (Mikolov et al., 2013) and GloVe (Pennington et al., 2014). However, the above models learn only a single context-independent word representation. To overcome this problem some models (Wieting et al., 2016; Bojanowski et al., 2017) enhance the representations with subword information, which can also somewhat deal with out-of-vocabulary words. Another approach is to learn a different representation for every word sense (Neelakantan et al., 2014) but this requires knowing the set of word senses in advance. More recent methods overcome the above issues by learning contextualized embeddings. Melamud et al. (2016) encode the context surrounding a pivot word using a bidirectional LSTM. Peters et al. (2018) use a deep bidirectional LSTM, learning word embeddings as functions of its internal states, calling the method Embeddings using Language Models (ELMo). We discuss ELMo in detail in Section 3. Devlin et al. (2018) introduced bidirectional encoder representations from transformers (BERT). This method learns pre-trained contextual embeddings by jointly conditioning on left and right context via an attention mechanism.

Program repair is an important task in software engineering and programming languages. For a detailed review see Monperrus (2018); Gazzola et al. (2019). Many recent program repair methods are based on machine learning. Yin et al. (2018) learn to represent code edits using a gated graph neural network (GGNN) (Li et al., 2016). Allamanis et al. (2018b) learn to identify a particular class of bugs called variable misuse bugs, using a GGNN. Chen et al. (2019) introduce SequenceR which learns to transform buggy lines into fixed ones via machine translation. Our work is orthogonal to these approaches and can be used as input in other models.

Finally, our work is also related to code representation methods many of which have also been used in program repair. Harer et al. (2018) learn Word2Vec embeddings for C/C++ tokens to predict software vulnerabilities. White et al. (2019) learn Word2Vec embeddings for Java tokens and utilize them in program repair. Alon et al. (2019) learn code embeddings using abstract syntax tree paths. A more detailed overview can be found in (Allamanis et al., 2018a; Chen & Monperrus, 2019).

## 3 EMBEDDINGS FROM LANGUAGE MODELS (ELMo)

ELMo (Peters et al., 2018) computes word embeddings from the hidden states of a language model. Consequently, the embeddings of each token depend on its context of the input sequence, even out-of-vocabulary (OOV) tokens have effective input representations. In this section, we briefly describe the ELMo embeddings.

The first step is that a neural language model is trained to maximize the likelihood of a training corpus. The architecture used by ELMo a bidirectional LSTM with $L$ layers and character convolutions in the input layer. Let the input be a sequence of tokens $(t_1, ...t_N)$. For each token $t_k$, denote by $\boldsymbol{x}_k^{LM}$ the input representation from the character convolution. Consequently, this representation passes through $L$ layers of forward and backward LSTMs. Then each layer $j \in \{1, ..., L\}$ of the forward LSTM computes a hidden state $\overrightarrow{\boldsymbol{h}_{k,j}^{LM}}$, and likewise the hidden states of the backward LSTM are denoted by $\overleftarrow{\boldsymbol{h}_{k,j}^{LM}}$. The parameters for the token representation and for the output softmax layer are tied for both directions, while different parameters are learned for each direction of the LSTMs.

After the language model has been trained, we can use it within another downstream task by combining the hidden states of the language model from each LSTM layer. This process is called ELMo. For each token $t_k$ of a sentence in the test set, the language model computes $2L + 1$ hidden states, one in each direction for each layer, and then the input layer. To make the following more compact, we can write these as $h_{k,0}^{LM} = x_k^{LM}$ for the input layer, and then $h_{k,j}^{LM} = [\overrightarrow{h_{k,j}^{LM}}, \overleftarrow{h_{k,j}^{LM}}]$ for all of the other layers. The set of these vectors is

$$R_k = \{h_{k,j}^{LM} | j = 0, ..., L\}. \tag{1}$$

To create the final representation that is fed to downstream tasks, ELMo collapses the set of representations into a single vector $E_k$ for token $t_k$. A simplistic approach is to only select the top layer, so that $E_k = h_{k,L}^{LM}$. A more general one, which we use in this work, is to combine the layers via fine-tuned task specific weights $\mathbf{s} = (s_1 \ldots s_L)$ for every

layer. Then we can compute the embedding for token $k$ as

$$E_k = \gamma \sum_{j=0}^{L} s_j h_{k,j}^{LM},$$ (2)

where $\gamma$ is an additional scalar parameter that scales the entire vector. In our experiments we did not performed fine-tuning and thus used equal weights $s_j = 1/(L+1)$ for each layer and $\gamma = 1$. However, our implementation also supports all the aforementioned ways of collapsing the set of representations.

A potential drawback of the method is that it still utilizes a softmax output layer with a fixed vocabulary that does not scale effectively and it still predicts UNK for OOV tokens which may have a negative effect on the representations.

## 4 SOURCE CODE ELMO

We describe Source Code ELMo (SCELMo), which trains ELMo on corpora of source code. However, we note that normally ELMo models in other domains are able to effectively utilize much larger representations. The code was tokenized using the esprima JavaScript tokenizer[1].   For training the ELMo model we used a corpus of 150,000 JavaScript Files (Raychev et al. 2016) consisting of various open-source projects. This corpus has previously been used on several tasks (Raychev et al., 2016; Pradel & Sen, 2018; Bavishi et al., 2018). We applied the patch released by Allamanis et al. (2018a) to filter out code duplication as this phenomenon was shown on this and other corpora to result in inflation of performance metrics. This resulted in 64750 training files and 33229 validation files. Since the validation set contains files from the same projects as the train the contained instances might be too similar and unrealistic overestimating. To address this we also created a test set of 500 random JavaScript projects sampled from the top 20,000 open-source JavaScript projects as of May 2019. The test corpus has not been previously utilized in previous work and is a better reflection of the performance of the learned bug detectors. Lastly, it is important to know what the performance of the method will be if we do not have access to training data from the projects on which we would like to find bugs. This is common in practice for many real case scenarios. For training the ELMo model, we use an embedding size of 100 features for each of the forward and backward LSTMs so that each layer sums up to 200 features.

## 5 CONTEXTUAL EMBEDDINGS FOR PROGRAM REPAIR

In this section, we describe how contextual embeddings can be incorporated within a recent machine learning-based bug detection system, the DeepBugs system of Pradel & Sen (2018). In the first part of this section, we give background about the DeepBugs system, and then we describe how we incorporate SCELMo within DeepBugs. DeepBugs treats the problem of finding a bug as a classification problem. The system considers a set of specific bug types, which are small mistakes that might be made in a program, such as swapping two arguments. For each bug type, DeepBugs trains a binary classifier that takes a program statement as input and predicts whether the statement contains that type of bug. At test time, this classifier can be run for every statement in the program to attempt to detect bugs.

In order to train the model both examples of correct and incorrect (buggy) code are necessary. DeepBugs treats the existing code as correct and randomly mutates it to obtain buggy code. To obtain training examples, we extract all expressions from the source code which are either the function calls with exactly two arguments and all binary expressions. To create instances of buggy code we mutate each of the correct instances. As such, arguments in function calls are swapped, the binary operator in binary expressions is replaced with another random one, and finally randomly either the left or the right operand is replaced by another random binary operand that appears in the same file. Then the classification task is a binary task to predict whether the instance is correct, i.e., it comes from the original code, or whether it is buggy, i.e. it was one of the randomly mutated examples. The validation and test sets are mutated in the same way as the training set. The split between correct and buggy instances has 50/50 class distribution as for each original code instance exactly one mutated buggy counterpart is created.

The architecture for the classifier is a feedforward network with a single hidden layer of 200 dimensions with Relu activations and a sigmoid output layer. For both the input and hidden layers a dropout of 0.2. The network was trained in all experiments for 10 epochs with a batch size of 50 and the RMSProp optimizer. We note that for maintaining a consistent comparison with DeepBugs we kept all the above parameters as well as the optimizer's parameters fixed to the values reported in Pradel & Sen (2018). Tuning these parameters would probably result in at least a small performance increase for our method.

---

[1]https://esprima.org/

```
1  // Argument order is inversed.
2  var delay = 1000;
3  setTimeout(delay, function() { // Function should be first.
4      logMessage(msgValue);
5  });
```

Listing 1: Swapped Arguments Bug

```
1  // && instead of || was used.
2  var p = new Promise();
3  if (promises === null && promises.length === 0) {
4    p.done(error, result);
5  }
```

Listing 2: Incorrect Binary Operator

```
1  // Call to .length is missing.
2  if ( index < matrix ) {
3    do_something();
4  }
```

Listing 3: Incorrect Binary Operand

Figure 1: Bug type examples.

In our experiments, we consider three bug types that address a set of common programming mistakes: swapped arguments of function calls, using the wrong binary operator and using an incorrect binary operand in a binary expression. The methodology can easily be applied to other bug types. Figure 1 illustrates an example of each of the three bug types.

### 5.1 INPUT TO THE CLASSIFIER

A key question is how a statement from the source code is converted into a feature vector that can be used within the classifier. DeepBugs uses a set of heuristics that, given a statement and a bug type, return a sequence of identifiers from the statement that are most likely to be relevant. For instance, for the call to setTimeout in Listing 1 the following sequence of identifiers would be extracted: *[setTimeout, delay, function]*. A detailed description of the heuristics is available in Appendix A.

These heuristics result in a sequence of program identifiers. These are converted to continuous vectors using word embeddings, concatenated, and this is the input to the classifier. DeepBugs uses Word2Vec embeddings trained on a corpus of code. In our experiments, we train classifiers using three different types of word embeddings. First, we kept the 10,000 most frequent identifiers/literals and assigned to each of them a *random embedding* of 200 features. Second, to reproduce the results of Pradel & Sen (2018), we use the CBOW variant of *Word2Vec* to learn representations consisting of 200 features for the 10,000 most frequent identifiers/literals. Finally, we train a *FastText* embeddings (Bojanowski et al., 2017) on the training set to learn identifier embeddings that contain subword information. The subwords used by FastText are all the character trigrams that appear in the training corpus. Identifiers are therefore composed of multiple subwords. To represent an identifier, we sum the embeddings of each of its subwords and summing them up. This allows the identifier embeddings to contain information about the structure and morphology of identifiers. This also allows the FastText embeddings, unlike the Word2Vec ones, to represent OOV words as a combination of character trigrams.

Note that DeepBugs can detect bugs only in statements that do not contain OOV (out-of-vocabulary) identifiers, because its Word2Vec embeddings cannot extract features for OOV names. Instead our implementation does not skip such instances. Since the original work discarded any instances that contain OOV identifiers we neither know how the method performs on such instances nor how often those appear in the utilized dataset of DeepBugs. Moreover, DeepBugs supported only a specific subset of AST nodes and skipped the rest. For example if a call's argument is a complex expression consisting of other expressions then the call would be skipped. However, we expanded the implementation to support all kinds of AST nodes and to not skip instances with nested expressions as discussed in Appendix A. We note that we still skip an instance if one of its main parts (e.g., a function call's argument) is a complex expression longer than 1,000 characters as such expressions might be overly long to reason about.

Table 1: Comparison of ELMo versus non-contextual embeddings for bug detection on a validation set of projects. Data is restricted to expressions that contain only single names.

|  | Random | Word2Vec | FastText | No-Context ELMo | SCELMo |
|---|---|---|---|---|---|
| Swapped Arguments | 86.18% | 87.38% | 89.55% | 90.02% | 92.11% |
| Wrong Binary Operator | 90.47% | 91.05% | 91.11% | 92.47% | 100.00% |
| Wrong Binary Operand | 75.56% | 77.06% | 79.74% | 81.71% | 84.23% |

## 5.2 CONNECTING SCELMO TO THE BUG DETECTOR

We investigated two variants of the bug detection model, which query SCELMo in different ways to get features for the classifier. The first utilizes the heuristic of Section A to extract a small set of identifiers or literals that represent the code piece. For example, for an incorrect binary operand instance we extract one identifier or literal for the left and right operands respectively, and we also extract its binary operator. Then, those are concatenated to form a query to the network. In the case of function calls we extract the identifier corresponding to the name of the called function, one identifier or literal for the first and second argument respectively and an identifiers for the expression on which the function is called. We also add the appropriate syntax tokens (a '.' if necessary, ',' between the two arguments, and left and right parentheses) to create a query that resembles a function call. This baseline approach creates simplistic fixed size queries for the network but does not utilize its full potential since the queries do not necessarily resemble actual code, nor correct code similar to the sequences in the training set for the embeddings. We will refer to this baseline as No-Context ELMo.

Our proposed method, we compute SCELMo embeddings to the language model all the tokens of the instances for which we need representations. Valid instances are functions calls that contain exactly two arguments and binary expressions. To create a fixed-size representation we extract only the features corresponding a fixed set of tokens. Specifically, for functions calls we use the representations corresponding to the first token of the expression on which the function is called, the function name, the first token of the first argument and the first token of the second argument. While, for binary expressions we use those of the first token of the left operand, the binary operator, and the first token of the right operand. Since the representations contain contextual information, the returned vectors can capture information about the rest of the tokens in the code sequence.

## 6 RESULTS

We next discuss the experiments we performed and their corresponding results. We measured the performance of the three baselines as well as those of non-contextual ELMO and SCELMO. Measuring the performance of non-contextual ELMO allows us to evaluate how much improvement is due to specifics of the language model architecture, such as the character convolutional layer which can handle OOVs, and how much is due to the contextual information itself.

### 6.1 PERFORMANCE ON VALIDATION SET

In our first experiment we evaluate the performance of the methods in tasks where training data from the same projects are available. The evaluation performed in this experiment gives a good estimation of how our method performs compared to the previous state-of-the-art technique of DeepBugs. One main difference however is that the evaluation now also includes instances which contain OOV. As a consequence the bug detections tasks are harder than those presented by Pradel & Sen (2018) as their evaluation does not include in both the training and validation set any instance for which an extracted identifier is OOV. Table 1 illustrates the performance of the baselines and our models. As one would expect the FastText baseline improves over Word2Vec for all bug types due to the subword information. Moreover, our model SCELMo massively outperforms all other methods. Lastly, even no-context ELMo the heuristic version of SCELMo that does not utilize contextual information at test time outperforms the baseline methods showcasing how powerful the pretrained representations are.

### 6.2 INCLUDING COMPLEX EXPRESSIONS

In our next experiment we also included instances that contain elements that are complex or nested expressions. For instance, in the original work if one the arguments of a function call or one of the operands of a binary expression is an expression consisting of other expressions then the instance would not be included in the dataset. Several AST node

Table 2: Comparison of SCELMo versus static embeddings on bug detection on a validation set of projects. Complex expressions are included in this validation set.

|  | Random | Word2Vec | FastText | No-Context ELMo | SCELMo |
|---|---|---|---|---|---|
| Swapped Arguments | 86.37% | 87.68% | 90.37% | 90.83% | 92.27% |
| Wrong Binary Operator | 91.12% | 91.68% | 91.92% | 92.75% | 100.00% |
| Wrong Binary Operand | 72.73% | 74.31% | 77.41% | 79.65% | 87.10% |

Table 3: Comparison of SCELMo versus static embeddings on bug detection on an external test set of 500 JavaScript projects.

|  | Random | Word2Vec | FastText | No-Context ELMo | SCELMo |
|---|---|---|---|---|---|
| Swapped Arguments | 75.79% | 78.22% | 79.40% | 81.37% | 84.25% |
| Wrong Binary Operator | 82.95% | 85.54% | 83.15% | 86.54% | 99.99% |
| Wrong Binary Operand | 67.46% | 69.50% | 72.55% | 75.74% | 83.59% |

types such as a `NewExpression` node or an `ObjectExpression` were not supported. Figure 2 a few examples of instances that would be previously skipped [2]. Such instances were skipped by Pradel & Sen (2018) and not included in their results. We do note though that we still skip very long expressions that contain more than 1000 tokens.

```
1  // First argument is binary expression
2  doComputation(x + find_min(components), callback);
```

```
1  // Second argument is an unsupported node
2  factory.test(simulator, new Car('Eagle', 'Talon TSi', 1993));
```

Figure 2: Examples of instances that would be skipped by DeepBugs.

Similarly to the previous experiment SCELMo significantly outperforms all other models. This is evident in Table 2. Lastly, we clarify that the results of this section should not be directly compared to those of the previous one as for this experiment the training set is also larger.

### 6.3 External Test Evaluation

The last experiment's objective is to showcase how the various models would perform on unseen projects as this better illustrates the generalizability of the techniques. The configuration utilized is identical to that of the previous section. By looking at Table 3 one can notice that the baselines have a major drop in performance. This is a common finding in machine learning models of code, namely, that applying a trained model to a new software project is much more difficult than to a new file in the same project. In contrast, SCELMo offers up to 15% improvement in accuracy compared to Word2Vec baseline. In fact, impressively enough SCELMo on the external test set is better than the evaluation set one of the baselines.

### 6.4 OOV Statistics

In order to better understand the above results we measured the OOV rate of the basic elements of the code instances appearing in the dataset. Here the OOV rate is calculated based on the vocabulary of 10000 entries utilized by the Word2Vec and random baseline models. These are illustrated in Tables 4 and 5. We measured the OOV rates for both the version of the dataset used in Section 6.4, which we call Train and Validation, and that used in Section 6.2, which we call Extended Train and Extended Validation.

Tables 4 and 5 describe the OOV rates for different parts of the expression types that are considered by the DeepBugs bug detector. A detailed description of the identifiers extraction heuristic can be found in Appendix A. We first focus

---

[2]The AST is extracted using the acorn parser `https://github.com/acornjs/acorn`

on the swapped arguments bug pattern and consider all of the method call that have exactly two arguments. Each method call contains the function name, a name of the first argument, a name of the second argument, and a base object. The base object is the identifier that would be extracted from the expression (if such an expression exists) on which the function is called. For instance, from the following expression: *window.navigator.userAgent.indexOf("Chrome")*, *userAgent* would be extracted as the base object. Table 4 shows for each of the components how often they are OOV. In the expanded version of the dataset if one of the arguments is a complex expression then it is converted into a name based on the heuristic described in Section A. The resulting statistics contain valuable information as for instance, it is almost impossible for the Word2Vec baseline to reason about a swap arguments bug if the identifiers extracted for both arguments are OOV.

In a similar manner for the incorrect operand and operator bug patterns we consider all the binary operations. Each binary expression consists of a left and right operand and a name is extracted for each of them. For each operand we also measured the frequency with which the operand corresponds to certain common types such as identifier, literal or a *ThisExpression*.

Table 4: OOV statistics for calls with exactly two arguments (Swapped arguments instances). The statistics are calculated on variants of the DeepBugs dataset.

|  | Train | Expanded Train | Validation | Expanded Validation |
|---|---|---|---|---|
| **Two Arguments Calls** | **574656** | **888526** | **289061** | **453486** |
| Calls Missing Base Object | 25.07% | 28.63% | 25.63% | 28.80% |
| Base Object Missing or OOV | 34.56% | 37.38% | 35.57% | 38.07% |
| Function Name OOV | 20.69% | 17.07% | 20.33% | 16.94% |
| First Argument OOV | 31.01% | 36.99% | 31.64% | 37.15% |
| Second Argument OOV | 27.25% | 22.86% | 27.94% | 23.49% |
| Both Arguments OOV | 11.33% | 9.57% | 11.96% | 10.16% |
| Base and Function Name OOV | 10.20% | 8.32% | 10.39% | 8.61% |
| Base and Arguments OOV | 4.21% | 3.31% | 4.88% | 3.77% |
| Function Name and Arguments OOV | 2.86% | 2.26% | 2.85% | 2.28% |
| All Elements OOV | 1.53% | 1.18% | 1.61% | 1.27% |

Table 5: OOV statistics for binary operations.

|  | Train | Expanded Train | Validation | Expanded Validation |
|---|---|---|---|---|
| **Binary Operations** | **1075175** | **1578776** | **540823** | **797108** |
| Left Operand OOV | 25.40% | 28.84% | 26.04% | 29.55% |
| Right Operand OOV | 20.37% | 23.98% | 20.74% | 24.55% |
| Both Operands OOV | 7.82% | 11.29% | 8.24% | 11.88% |
| Unknown Left Operand Type | 83.36% | 87.80% | 83.14% | 87.74% |
| Unknown Right Operand Type | 48.48% | 47.23% | 48.47% | 47.05% |
| Both Operand Types Unknown | 33.34% | 36.06% | 33.20% | 35.87% |
| All OOV or Unknown | 3.59% | 4.03% | 3.81% | 4.3% |

## 7    IS NEURAL BUG-FINDING USEFUL IN PRACTICE?

Although related work (Pradel & Sen, 2018; Allamanis et al., 2018b; Vasic et al., 2019) has shown that there is great potential for embedding based neural bug finders, the evaluation has mostly focused on synthetic bugs introduced by mutating the original code. However, there is no strong indication that the synthetic bugs correlate to real ones, apart from a small study of the top 50 warnings for each bug type produced by DeepBugs. A good example is the mutation operation utilized for the incorrect binary operator bug. A lot of the introduced bug instances could result in syntactic errors. This can potentially create a classifier with a high bias towards correlating buggy code to syntactically incorrect code, thus hindering the model's ability to generalize on real bugs. Ideally, in an industrial environment we would like the resulting models to achieve a false positive rate of less than 10 % (Sadowski et al., 2015). Sadly, high true positive rates are not to be expected as well since static bug detectors were shown to be able to detect less than 5% of bugs

Table 6: Real bug mined instances.

|  | Swapped Arguments | Wrong Binary Operator | Wrong Binary Operand |
|---|---|---|---|
| Mined Instances | 303 | 80 | 1007 |

Table 7: Real bug identification task recall and false positive rate (FPR).

|  | Word2Vec-Recall | Word2Vec-FPR | SCELMo-Recall | SCELMo-FPR |
|---|---|---|---|---|
| Swapped Arguments | 3.34% | 0.33% | 49.67% | 33.78% |
| Wrong Binary Operator | 8.95% | 7.70% | 0.00% | 0.00% |
| Wrong Binary Operand | 11.99% | 12.11% | 15.81% | 14.34% |

(Habib & Pradel, 2018) contained in the Defects4J corpus (Just et al., 2014) and less than 12% in a single-statement bugs corpus (Karampatsis & Sutton, 2019). We note that in the second case the static analysis tool is given credit by reported any warning for the buggy line, so the actual percentage might lower than the reported one.

We next make a first step on investigating the practical usefulness of our methods by applying the classifiers of the previous section on a small corpus of real JavaScript bugs. However, we think that this is a very hard yet interesting problem that should be carefully examined in future work. In order to mine a corpus of real bug changes we used the methodology described in (Karampatsis & Sutton, 2019). We note that we adapted their implementation to utilize the Rhino JavaScript parser[3]. Their methodology extracts bug fixing commits and filters them to only keep those that contain small single-statement changes. Finally, it classifies each pair of modified statements by whether the fit a set of mutation patterns. The resulting dataset is shown in Table 6. Upon acceptance of the paper we will release this dataset along with our implementation, the rest of data used, and the learned representations.

Finally, we queried the DeepBugs and SCELMo with each buggy instance as well as its fixed variant and measured the percentage of correctly classified instances for each of the two categories. We also ignored any instances for which the JavaScript parser utilized for both failed to extract an AST. We classified as bugs any instances that were assigned a probability to be a bug $> 75\%$. In an actual system this threshold should ideally be tuned on a validation set.

Table 7 suggests that there might indeed be some potential for future practical applications of neural bug finding techniques. Both are able to uncover some of the bugs. However, the results also suggest that careful tuning of the predictions threshold might be necessary, especially if we take into account the industrial need to comply with a low false positive rate (FPR). For instance, raising SCELMo's prediction threshold to $80\%$ for the swap arguments bug results in finding only 3.34% of the bugs but correctly classifying 100% of the repaired function calls, thus achieving 0.0% false positive rate. Moreover, since SCELMo could not uncover any of the real binary operator bugs, future work could investigate the effect of utilizing different mutation strategies for the purpose of artificial bug-induction. Future work could also investigate if fine-tuning on small set of real bugs could result in more robust classifiers.

## 8   CONCLUSION

We have presented SCELMo, which is to our knowledge the first language-model based contextual embeddings for source code. Contextual embeddings have many potential advantages for source code, because surrounding tokens can indirectly provide information about tokens, e.g. about likely values of variables. We highlight the utility of SCELMo embeddings by using them within a recent state-of-the-art machine learning based bug detector. The SCELMo embeddings yield a dramatic improvement in the synthetic bug detection performance benchmark, especially on lines of code that contain out-of-vocabulary tokens and complex expressions that can cause difficulty for the method. We also showed and discussed the performance of the resulting bug detectors on a dataset of real bugs raising useful insights for future work.

---

[3] https://github.com/mozilla/rhino

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

## A    NAME EXTRACTION HEURISTIC

In order for DeepBugs to operate it is necessary to extract identifiers or literals for each expression part of the statement. The bug detector for swapped arguments utilizes the following elements of the function call:

**Base Object:**  The expression on which the function is called.

**Callee:**  The called function.

**Argument 1:**  The expression consisting the first argument of the called function.

**Argument 2:**  The expression consisting the first argument of the called function.

Similarly the bug detectors for incorrect binary operators and operands utilize the following elements of the binary expression:

**Binary Operator:**  The binary operator utilized in the expression.

**Left Operand:**  The left operand of the binary expression.

**Right Operand:**  The right operand of the binary expression.

We next describe the extraction heuristic, which is shared by all the bug detectors. The heuristic takes as input a node $n$ representing an expression and returns $name(n)$ based on the following rules:

- Identifier: return its name.
- Literal: return its value.
- this expression: return *this*.
- Update expression with argument $x$: return $name(x)$.
- Member expression accessing a property $p$: return $name(p)$.
- Member expression accessing a property $base[p]$: return $name(base)$.
- Call expression $base.callee(...)$: return $name(callee)$.
- Property node $n$: If $n.key$ does not exist return $name(n.value)$. If $name(n.key)$ does not exist return $name(n.value)$ . Otherwise randomly return either $name(n.value)$ or $name()n.key)$.
- Binary expression with left operand $l$ and right operand $r$: Run the heuristic on both $l$ and $r$ to retrieve $name(l)$ and $name(r)$. If $name(l)$ does not exist return $name(r)$. If $name(r)$ does not exist return $name(l)$. Otherwise randomly return either $name(l)$ ir $name(r)$.
- Logical expression with left operand $l$ and right operand $r$: Run the heuristic on both $l$ and $r$ to retrieve $name(l)$ and $name(r)$. If $name(l)$ does not exist return $name(r)$. If $name(r)$ does not exist return $name(l)$. Otherwise randomly return either $name(l)$ ir $name(r)$.
- Assignment expression with left operand $l$ and right operand $r$: Run the heuristic on both $l$ and $r$ to retrieve $name(l)$ and $name(r)$. If $name(l)$ does not exist return $name(r)$. If $name(r)$ does not exist return $name(l)$. Otherwise, randomly return either $name(l)$ ir $name(r)$.
- Unary expression with argument $u$ : Return $name(u)$.
- Array expression with elements $l_i$ : For all $l_i$ that $name(l_i)$ exists randomly choose one of them and return $name(l_i)$.
- Conditional expression with operands $c$, $l$, and $r$: Randomly choose one out of $c, l, r$ for which a name exists and return its name.
- Function expression: return $function$.
- Object expression: return {.
- New expression with a constructor function call $c$: return $name(c)$.

All random decisions follow a uniform distribution.

