# OpenReview forum: "SCELMo: Source Code Embeddings from Language Models"
_ICLR.cc/2020/Conference — Reject_

### Official Review · AnonReviewer1 · 2019-10-23
**Official Blind Review #1**

**Rating:** 3

**Review:**

The paper proposes to use ELMO embeddings to improve the precision on the first step of the DeepBugs tasks defined by Pradel and Sen (2018). This first step is an artificial problem created by taking real programs (with and without bugs, but assuming almost all of them are correct) and introducing bugs of certain type into the programs. Then, a classifier needs to distinguish between the real and the artificial set. This classifier is then to be used as a checker for anomalies in code and the anomalies are reported as bugs, however the paper skips this second step and only reports results on the first classification problem.

Technically, the paper improve this first step of DeepBugs by using a standard variant of ELMO. The evaluation is detailed, but the results are unsurprising. The paper simply tech-transfers the idea from NLP to Code. If this work is accepted at the conference, I cannot imagine an interesting presentation or a poster that simply cites the changed numbers. Did we expect ELMO to be worse than more naive or random embeddings?

The work and its results heavily peg on the DeepBugs and increases the precision of its first step by a significant margin, but does not show getting any more useful results.  In fact, on one task (Wrong Binary Operator), SCELmo gets to 100% accuracy. This means it will never report any bugs, whereas DeepBugs seems to be performing best on exactly this kind of reports with its weaker model.

I would recommend the authors to either work on showing practical usefulness of the technique, showing something for the full bugfinding task (not merely the first, artificial part), or to investigate if (or how) the idea to add bug-introducing changes to a code dataset is conceptually flawed for bugfinding (as this idea is widely used by several other works like Allamanis et al 2018b or by https://arxiv.org/abs/1904.01720 which also don't get to practical tools ). There seems to be some indication of this by the reported 100% accuracy, but right now this remains completely uninvestigated.

Minor issues:
Listing 3: Opernad -> Operand
Page 5. There is no Table 6.1


**Experience Assessment:**

I have published one or two papers in this area.

**Review Assessment: Checking Correctness Of Derivations And Theory:**

I carefully checked the derivations and theory.

**Review Assessment: Checking Correctness Of Experiments:**

I carefully checked the experiments.

**Review Assessment: Thoroughness In Paper Reading:**

I read the paper thoroughly.

---

> ### Author Response · Authors · 2019-11-12
> **Response to Reviewer #1**
>
> Thank you for the feedback and your insightful comments.
>
> Although the results might be somewhat unsurprising we believe that this work can offer empirical evidence for the effectiveness of this kind of techniques in a new domain, for which there is neither empirical evidence nor pretrained models. We also offer insight in the paper why these techniques would be a good fit for it.
>
> The method can still report bugs in the code even when it achieves 100% accuracy in the synthetic evaluation, because we can still rank code locations by the probability that they contain a bug type, thus obtaining a ranked list of the most suspicious locations in unseen code. Also, we know from other work that this particular bug type (Wrong Binary Operator) is actually fairly rare in practice (we keep this in the evaluation to compare to DeepBugs), so it is not that surprising that the classifier does not identify clear instances of the bugs.
>
>  Furthermore, we cannot make the strong assumption that misclassifying more instances means that we’ll find more real bugs as there is no guarantee that the misclassified locations are indeed bugs. Especially, since for industrial tools such as Google’s Tricorder (Sadowski, 2015) a false positive rate of less than 10% is enforced.  As a consequence in an industrial setting it would be prefered to use bug detectors with high precision as this will result in more trustworthy tools for the developers due to less overhead.
>
> We also think that showcasing the practical usefulness of the technique and exploring whether the idea of bug-introducing changes is effective in practice is a very good idea. We will look into this.
>
> We will fix all minor issues.

---

### Official Review · AnonReviewer3 · 2019-10-23
**Official Blind Review #3**

**Rating:** 3

**Review:**

This paper leverage recent advances of ELMo in context embedding and apply it in the source code embedding. With the help of ELMo, source embedding can take the three benefits: (1)  Surrounding names provide indirect information about possible values the variable could take; (2) an variable’s value evolves through the program execution can be captured; (3) open a gate for the reuse of the ptr-trained model. To evaluate the effectiveness of the proposed approach, authors conduct experiments on the downstream task of the bug detection.
Pros:
1. This work study an interesting problem, which is challenging to solve.
2. The application and combination of different techniques in this paper are smart.
3. The experiment results show better performance of contextual embedding based method compared with non-contextual embedding based methods.
Cons:
1. It is a good application of known techniques, but the novelty is limited.
2. It is suggested to evaluate the effectiveness of the proposed approach on various source code analysis task such as variable misuse.
3. It is suggested to compare with other state-of-the-art baseline methods, e.g. BERT.
4. In the end of the introduction section, the authors claim that "we release our implementation and representation...". However, implementation, representation and dataset are missing.

**Experience Assessment:**

I have read many papers in this area.

**Review Assessment: Checking Correctness Of Derivations And Theory:**

I carefully checked the derivations and theory.

**Review Assessment: Checking Correctness Of Experiments:**

I carefully checked the experiments.

**Review Assessment: Thoroughness In Paper Reading:**

I read the paper thoroughly.

---

> ### Author Response · Authors · 2019-11-12
> **Response to Reviewer #3**
>
> Thank you for the feedback and your insightful comments.
> Regarding the issues that you highlight.
> 1. The novelty in this work comes from exploring whether contextual embeddings would be effective in this new domain, for which similar techniques have not been applied in the literature.
> 2. This is a good suggestion. We will definitely take it into account for future work.
> 3. Although this is a reasonable thought, we are not aware of pre-trained BERT embeddings in the literature. Unfortunately, training from scratch a BERT model in academia is currently infeasible.
> 4. The training and validation data are available. We will release the test data through an institutional repository DOI. In order to not break anonymity we did not include this to the current version of the paper. The code is already in a private GitHub repository, we’ll make it public upon acceptance.

---

### Official Review · AnonReviewer2 · 2019-10-25
**Official Blind Review #2**

**Rating:** 8

**Review:**

The paper proposes an embedding method for source code tokens, which is based on contextual word representation, particularly is based on the method of ELMo. The learned representation is evaluated on the task of bug detection, with promising performance.

Strengths:
The paper addresses an important and impactful problem. The solution designed for this problem seems very reasonable. Experiments are useful and reasonable and the experimental results are promising and in the favor of the paper.
The paper is well written and clear.

Weaknesses:
- The data used (in particular the method of buggy code generation applied) seems very specific.  It would be interesting to know the performance of the method on real bugs.
- The paper is a bit low in technicality.

Decision: Accept
I think this paper is overall a good work and can open direction of research even beyond the scope of the paper, for example  in combining learning and reasoning, or in source code generation with adversarial models.

Minor:
- Since compilers can spot errors in code completely, it would be useful to motivate the advantage of learning for bug detection
- The table referrals in the body of the paper contains wrong table numbers in Sections 6.1, 6.2, 6.3.
- The incorrect Binary Operator example in Listing 2 does not seem to be a well justified bug. It could be a correct piece of code for a different purpose.
- which use -> which we use

**Experience Assessment:**

I have published one or two papers in this area.

**Review Assessment: Checking Correctness Of Derivations And Theory:**

N/A

**Review Assessment: Checking Correctness Of Experiments:**

I assessed the sensibility of the experiments.

**Review Assessment: Thoroughness In Paper Reading:**

I read the paper thoroughly.

---

> ### Author Response · Authors · 2019-11-12
> **Response to Reviewer #2**
>
> Thank you for the feedback and your insightful comments.
> Evaluating the performance of the method on real bugs is a great suggestion. We will look into this.
> Compilers are indeed great at spotting syntactic errors the proposed approach can go beyond that and detect semantic errors that a compiler would be unable to. We’ll make that more clear in the paper.
> We will fix the table indices and update Listing 2.

---

### Author Response · Authors · 2019-11-15
**Paper Updated**

We would like to thank all reviewers for their feedback and insightful comments.

We would like to inform the reviewers that we have revised our submission to include a new section where we discuss whether the idea to add bug-introducing changes to a code dataset has practical usefulness for bug-finding. In the same section we also measure performance on a small dataset of real bugs, which we mined. We would be grateful if you could take a look at this and consider whether this improves your judgement about this submission.

---

### Decision · Program_Chairs · 2019-12-19

**Decision:**

Reject

**Comment:**

This paper improves DeepBugs by borrowing the NLP method ELMo as new representations. The effectiveness of the embedding is investigated using the downstream task of bug detection.

Two reviewers reject the paper for two main concerns:
1 The novelty of the paper is not strong enough for ICLR as this paper mainly uses a standard context embedding technique from NLP.
2 The experimental results are not convincing enough and more comprehensive evaluation are needed.

Overall, this novelty of this paper does not meet the standard of ICLR.